# Symmetry-based indicators of band topology in the 230 space groups

Hoi Chun Po[1,2], Ashvin Vishwanath[1,2] & Haruki Watanabe[3]

The interplay between symmetry and topology leads to a rich variety of electronic topological phases, protecting states such as the topological insulators and Dirac semimetals. Previous results, like the Fu-Kane parity criterion for inversion-symmetric topological insulators, demonstrate that symmetry labels can sometimes unambiguously indicate underlying band topology. Here we develop a systematic approach to expose all such symmetry-based indicators of band topology in all the 230 space groups. This is achieved by first developing an efficient way to represent band structures in terms of elementary basis states, and then isolating the topological ones by removing the subset of atomic insulators, defined by the existence of localized symmetric Wannier functions. Aside from encompassing all earlier results on such indicators, including in particular the notion of filling-enforced quantum band insulators, our theory identifies symmetry settings with previously hidden forms of band topology, and can be applied to the search for topological materials.

[1] Department of Physics, University of California, Berkeley, CA 94720, USA. [2] Department of Physics, Harvard University, Cambridge, MA 02138, USA. [3] Department of Applied Physics, University of Tokyo, Tokyo 113-8656, Japan. Correspondence and requests for materials should be addressed to A.V. (email: avishwanath@g.harvard.edu)

The discovery of topological insulators (TIs) has reinvigorated the well-established theory of electronic band structures (BSs)[1, 2]. Exploration along this dimension has led to an ever-growing arsenal of topological materials, which include, for instance, topological (crystalline) insulators[3–5], quantum anomalous Hall insulators[6] and Weyl and Dirac semimetals[7]. Such materials possess unprecedented physical properties, like quantized response and gapless surface states, that are robust against all symmetry-preserving perturbations as long as a band picture remains valid[1, 2, 7].

Soon after these developments, it was realized that symmetries of energy bands, a thoroughly studied aspect of band theory, is also profoundly intertwined with topology. This is exemplified by the celebrated Fu-Kane criterion for inversion-symmetric materials, which demarcates TIs from trivial insulators using only their parity eigenvalues[8]. This criterion, when applicable, greatly simplifies the topological analysis of real materials, and underpins the theoretical prediction and subsequent experimental verification of many TIs[8–12].

It is of fundamental interest to obtain results akin to the Fu-Kane criterion in other symmetry settings. Early generalizations in systems without time-reversal (TR) symmetry in two-dimensional (2D) constrained the Chern number ($C$). The eigenvalues of an $n$-fold rotation were found to determine $C$ modulo $n$[13–15]. This is characteristic of a symmetry-based indicator of topology—when the indicator is nonvanishing, band topology is guaranteed, but certain topological phases (i.e., $C$ a multiple of $n$ in this context) may be invisible to the indicator. In three-dimensional (3D) systems, it was also recognized that spatial inversion alone can protect nontrivial phases. A feature here is that these phases do not host protected surface states, since inversion symmetry is broken at the surface, but they do represent distinct phases of matter. For example, they possess nontrivial Berry phase structure in the Brillouin zone, which leads to robust entanglement signatures[13, 14, 16, 17] and, in some cases, quantized responses[13, 14, 18]. Interestingly, in the absence of TR invariance the inversion eigenvalues can also protect Weyl semimetals[13, 14], which informed early work on materials candidates[19]. Hence, these symmetry-based indicators are relevant both to the search for nontrivial insulating phases, and also to the study of topological semimetals. It is also important to note that the goal here is to identify signatures of band topology in the symmetry transformations of the state, which is distinct from the full classification of topological phases.

An important open problem is to extend these powerful symmetry indicators for band topology to all space groups (SGs). Earlier studies have emphasized the topological perspective, which typically rely on constructions that are specifically tailored to particular band topology of interest[8, 15, 20, 21]. While some general mathematical frameworks have been developed[22–24], obtaining a full list of concrete results from such an approach faces an inherent challenge stemming from the sheer multitude of physically relevant symmetry settings—there are 230 SGs in 3D, and each of them is further enriched by the presence or absence of both spin–orbit coupling and TR symmetry.

A complementary, symmetry-focused perspective leverages the existing exhaustive results on band symmetries[25, 26] to simplify the analysis. Previous work along these lines has covered restricted cases[13, 14, 27, 28]. For instance, in ref. [28], which focuses on systems in the wallpaper groups without any additional symmetry, such an approach was adopted to help develop a more physical understanding of the mathematical treatment of ref. [22]. However, the notion of nontriviality is a relative concept in these approaches. While such formulation is well-suited for the study of phase transitions between different systems in the same symmetry setting, it does not always indicate the presence of underlying band topology. As an extreme example, such classifications generally regard atomic insulators (AIs) with different electron fillings as distinct phases, although all the underlying BSs are topologically trivial.

Here, we adopt a symmetry-based approach that focuses on probing the underlying band topology. At the crux of our analysis is the observation that topological BSs arise whenever there is a mismatch between momentum-space and real-space solutions to symmetry constraints[29, 30]. To quantitatively expose such mismatches, we first develop a mathematical framework to efficiently analyze all possible BSs consistent with any symmetry setting, and then discuss how to identify the subset of BSs arising from AIs, which are formed by localizing electrons to definite orbitals in real space. The mentioned mismatch then follows naturally as the quotient between the allowed BSs and those arising from real-space specification. We compute this quotient for all 230 SGs with or without spin–orbit coupling and/or TR symmetry. Using these results, we highlight symmetry settings suitable for finding topological materials, including both insulators and semimetals. In particular, we will point out that, in the presence of inversion symmetry, stacking two strong 3D TIs will not simply result in a trivial phase, despite all the $\mathbb{Z}_2$ indices have been trivialized. Instead, it is shown to produce a quantum band insulator (QBI)[30], which can be diagnosed through its robust gapless entanglement spectrum.

## Results

**Overview of strategy and results.** Our major goal is to systematically quantify the mismatch between momentum-space and real-space solutions to symmetry constraints in free-electron problems[30]. While AIs, which by definition possess localized symmetric Wannier orbitals, can be understood from a real-space picture with electrons occupying definite positions as if they were classical particles, topological BSs (that are intrinsic to dimensions greater than one) do not admit such a description. Whenever there is an obstruction to such a real-space reinterpretation for a band insulator, the insulating behavior can only be understood through the quantum interference of electrons, and we refer to such systems as QBIs. While all topological phases such as Chern insulators, weak and strong $\mathbb{Z}_2$ TIs and topological crystalline insulators with protected surface states in $d > 1$ are QBIs, more generally, QBIs may not have nontrivial surface states when the protecting symmetries are not compatible with any surface termination. Nonetheless, they represent distinct phases of matter and showcase nontrivial Berry phase in the Brillouin zone[31], robust entanglement signatures[13, 14, 16, 30], and sometimes quantized responses[13, 14, 18].

Building on this insight, we develop an efficient strategy for identifying topological materials indicated by symmetries. We will first outline a simple framework to organize the set of all possible BSs using only their symmetry labels. By extending the ideas in refs [13, 28] and allowing for both addition (stacking) as well as formal subtractions of bands, we show that BSs can be conveniently represented in terms of a special type of Abelian group, which is simply called a lattice in mathematical nomenclature. Next, to isolate topological BSs we quotient out those that can arise from a Wannier description. Since such band topology is uncovered from the symmetry representations of the bands, we will refer to it as being represented-enforced. In this work, we present the results of this computation for all of the 230 SGs, 80 layer groups, and 75 rod groups, covering all cases with or without TR symmetry and spin–orbit coupling. Our scheme automatically encompasses all previous results concerning symmetry indicators of band topology, including in particular the Fu-Kane criterion, the relation between Chern numbers and

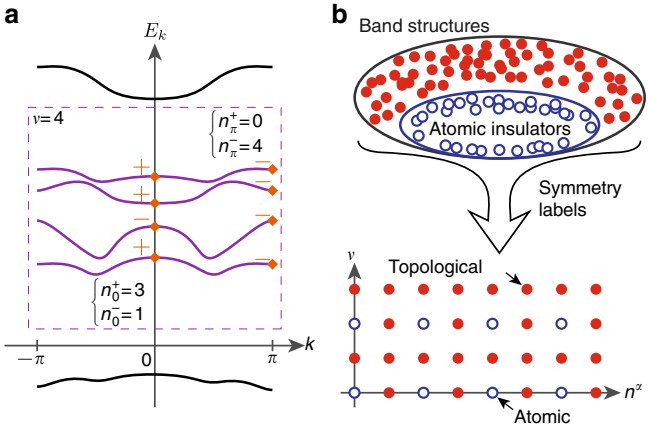

**Fig. 1** Symmetry-based indicators of band topology. **a** Symmetry labeling of bands in a 1D inversion-symmetric example. $k_0 = 0, \pi$ are high-symmetry momenta, where the bands are either even (+) or odd (−) under inversion symmetry (*orange diamonds*). From a symmetry perspective, a target set of bands (*purple* and *boxed*) separated from all others by band gaps can be labelled by the multiplicities of the two possible symmetry representations, which we denote by the integers $n_{k_0}^{\pm}$. Note that such labeling is insensitive to the detailed energetics within the set. In addition, the set is also characterized by the number of bands involved, which we denote by $\nu$. Altogether, the set is characterized by five integers, which are further subjected to the constraints $\nu = n_0^+ + n_0^- = n_\pi^+ + n_\pi^-$. **b** Symmetry labels like those described in **a** can be similarly defined for systems symmetric under any of the 230 space groups in three dimensions. Using such labels, one can reinterpret the set of band structures as an Abelian group. This is schematically demonstrated through the two labels $\nu$ and $n^\alpha$, which organize the set of all possible band structures into a two-dimensional lattice. Note that the dimensionality of this lattice is given by the number of independent symmetry labels, and is a property of the symmetry setting at hand. Organized this way, the band structures corresponding to atomic insulators, which are trivial by our definition, will generally occupy a sublattice. Any band structure that does not fall within this sublattice necessarily possesses nontrivial band topology

rotation eigenvalues, and the inversion-protected nontrivial phases.

We will utilize the results and identify representation-enforced QBIs (reQBIs). We will also discuss a more constrained approach where one first specifies the microscopic lattice degrees of freedom. This is relevant to materials where a hierarchy of energy scales isolates a group of atomic orbitals. We find examples where these constraints lead to semimetallic behavior, despite band insulators at the same filling are symmetry-allowed. We will refer to these as lattice-enforced semimetals (leSMs) and give a concrete tight-binding example of them. Generalizations of these approaches should aid in the discovery of experimentally relevant topological semimetals and insulators.

Finally, we make two remarks. First, we ignore electron–electron interactions. Second, while our approach is applicable in any dimension, in the special case of one-dimensional (1D) problems even topological phases are smoothly connected to AIs[32], and therefore are regarded as trivial within our framework. These states, and their descendants in higher dimensions, are collectively known as frozen-polarization insulators[13], and will be absent from our discussion on topological phases.

**BSs form an Abelian group**. Here, we argue that the possible set of BSs symmetric under an SG $\mathcal{G}$ can be naturally identified as the group $\mathbb{Z}^{d_{BS}} \equiv \mathbb{Z} \times \mathbb{Z} \times \ldots \times \mathbb{Z}$, where $d_{BS}$ is a positive integer that

depends on both $\mathcal{G}$ and the spin of the particles (Fig. 1). We will first set aside TR symmetry, and later discuss how it can be easily incorporated into the same framework. The discussion in this section follows immediately from well-established results concerning band symmetries[25], and the same set of results was recently utilized in ref. [28] to discuss an alternative way to understand the more formal classification in ref. [22]. Although there is some overlap between the discussion here and that in ref. [28], we will focus on a different aspect of the narration: instead of being solely concerned with the values of $d_{BS}$, we will be more concerned with utilizing this framework to extract other physical information about the systems.

We begin by reviewing some basic notions using a simple example. Consider free electrons in a 1D, inversion-symmetric crystal. The energy bands $E_m(k)$ are naturally labeled by the band index $m$ and the crystal momentum $k \in (-\pi, \pi)$. Since inversion $P$ flips $k \leftrightarrow -k$, the Bloch Hamiltonian $H(k)$ is symmetric under $PH(k)P^{-1} = H(-k)$, which implies $E_m(k) = E_m(-k)$, and the wavefunctions are similarly related. The two momenta $k_0 = 0$ and $\pi$ are special as they satisfy $P(k_0) = k_0$ (up to a reciprocal lattice vector). As such, the symmetry constraint imposed by $P$ becomes a local constraint at $k_0$, which implies the wavefunctions $\psi_m(k_0)$ (generically) furnish irreducible representations (irreps) of $P$: $\psi_m^\dagger(k_0) P \psi_m(k_0) = \zeta_m(k_0)$, with $\zeta_m(k_0) = \pm 1$.

The parities $\zeta_m(k_0) = \pm 1$ can be regarded as local (in momentum space) symmetry labels for the energy band $E_m(k)$, and such labels can be readily lifted to a global one assigned to any set of bands separated from others by a band gap. We will refer to such sets of bands as BSs, although, as we will explain, caution has to be taken when this notion is used in higher dimensions. Insofar as symmetries are concerned, we can label the BS by its filling, $\nu$, together with the four non-negative integers, $n_{k_0}^\pm$, corresponding to the multiplicity of the irrep $\pm$ at $k_0$ (Fig. 1a). Generally, such labels are not independent, since the assumption of a band gap, together with the continuity of the energy bands, casts global symmetry constraints on the symmetry labels. These constraints are known as compatibility relations. For our 1D problem at hand, there are only two of them, which arise from the filling condition: $\nu = n_0^+ + n_0^- = n_\pi^+ + n_\pi^-$. Consequently, the BS is fully specified by three non-negative integers, which we can choose to be $n_0^+$, $n_\pi^+$, and $\nu$.

This discussion to this point is similar to that of ref. [28], but we now depart from the combinatorics point of view of that work. Instead, similar to ref. [13] we develop a mathematical framework to efficiently characterize energy bands in terms of their symmetry transformation properties, and then show that it provides a powerful tool for analyzing general BSs. To begin, we first note that any BS in this 1D inversion-symmetry problem can be represented by a five-component "vector" $\mathbf{n} \equiv (n_0^+, n_0^-, n_\pi^+, n_\pi^-, \nu) \in \mathbb{Z}_{\geq 0}^5$, where $\mathbb{Z}_{\geq 0}$ denotes the set of non-negative integers. In addition, $\mathbf{n}$ is subjected to the two compatibility relations. We can arrange these relations into a system of linear equations and denote them by a $2 \times 5$ matrix $\mathcal{C}$. The admissible BSs then satisfy $\mathcal{C}\mathbf{n} = 0$, and hence $\ker \mathcal{C}$, the solution space of $\mathcal{C}$, naturally enters the discussion. For the current problem, $\ker \mathcal{C}$ is 3D, which echoes with the claim that the BS is specified by three non-negative integers.

At this point, it is natural to make a mathematical abstraction and lift the physical condition of non-negativity. We define

$$\{BS\} \equiv \ker \mathcal{C} \cap \mathbb{Z}^D, \tag{1}$$

where for the 1D problem at hand we have $D = 5$. The main advantage of this abstraction is that, unlike $\mathbb{Z}_{\geq 0}^D$, $\mathbb{Z}^D$ is an Abelian group, which greatly simplifies our forthcoming analysis.

**Table 1 Characterization of band structures for systems with time-reversal symmetry and significant spin–orbit coupling**

| d | Space groups |
|---|---|
| 1 | 1, 3, 4, 5, 6, 7, 8, 9, 16, 17, 18, 19, 20, 21, 22, 23, 24, 25, 26, 27, 28, 29, 30, 31, 32, 33, 34, 35, 36, 37, 38, 39, 40, 41, 42, 43, 44, 45, 46, 76, 77, 78, 80, 91, 92, 93, 94, 95, 96, 98, 101, 102, 105, 106, 109, 110, 144, 145, 151, 152, 153, 154, 169, 170, 171, 172, 178, 179, 180, 181 |
| 2 | 79, 90, 97, 100, 104, 107, 108, 146, 155, 160, 161, 195, 196, 197, 198, 199, 208, 210, 212, 213, 214 |
| 3 | 48, 50, 52, 54, 56, 57, 59, 60, 61, 62, 68, 70, 73, 75, 89, 99, 103, 112, 113, 114, 116, 117, 118, 120, 122, 133, 142, 150, 157, 159, 173, 182, 185, 186, 209, 211 |
| 4 | 63, 64, 72, 121, 126, 130, 135, 137, 138, 143, 149, 156, 158, 168, 177, 183, 184, 207, 218, 219, 220 |
| 5 | 11, 13, 14, 15, 49, 51, 53, 55, 58, 66, 67, 74, 81, 82, 86, 88, 111, 115, 119, 134, 136, 141, 167, 217, 228, 230 |
| 6 | 69, 71, 85, 125, 129, 132, 163, 165, 190, 201, 203, 205, 206, 215, 216, 222 |
| 7 | 12, 65, 84, 128, 131, 140, 188, 189, 202, 204, 223 |
| 8 | 124, 127, 148, 166, 193, 200, 224, 226, 227 |
| 9 | 2, 10, 47, 87, 139, 147, 162, 164, 176, 192, 194 |
| 10 | 174, 187 |
| 11 | 225, 229 |
| 13 | 83, 123 |
| 14 | 175, 191, 221 |

*d* the rank of the Abelian group formed by the set of band structures

In particular, {BS} so defined can be identified with $\mathbb{Z}^{d_{BS}}$, where $d_{BS} = 3$ is the dimension of the solution space $\ker \mathcal{C}$. Physically, the addition in $\mathbb{Z}^{d_{BS}}$ corresponds to the stacking of energy bands.

Next, we generalize the discussion to any SG $\mathcal{G}$ in three dimensions. We call a momentum **k** a high-symmetry momentum if there is any $g \in \mathcal{G}$ other than the lattice translations such that $g(\mathbf{k}) = \mathbf{k}$ (up to a reciprocal lattice vector). We define a BS as a set of energy bands isolated from all others by band gaps above and below at all high-symmetry momenta. Note that, in 3D, the phrase "all high-symmetry momenta" includes all high-symmetry points, lines, and planes. The discussion for the 1D example carries through, except that one has to consider a much larger zoo of irreps and compatibility relations[25] (see Methods and Supplementary Notes 1 and 2 for a detailed discussion).

While Eq. 5 follows readily from definitions, it has interesting physical implications. As a group, $\mathbb{Z}^{d_{BS}}$ is generated by $d_{BS}$ independent generators. In the additive notation, natural for an Abelian group, we can write the generators as $\{\mathbf{b}_i : i = 1, \ldots, d_{BS}\}$, and for any given BS we can expand it similar to elements in a vector space

$$\text{BS} = \sum_{i=1}^{d_{BS}} m_i \mathbf{b}_i, \tag{2}$$

where $m_i \in \mathbb{Z}$ are uniquely determined once the basis is fixed. Therefore, full knowledge of {BS} is obtained once the $d_{BS}$ generators $\mathbf{b}_i$ are found.

So far, we have not addressed the effect of TR symmetry, which, being anti-unitary, does not lead to new irreps when it is incorporated[25]. Instead, TR symmetry could force certain irreps to become paired with either itself or another, giving rise to additional constraints on **n**. Nonetheless, these constraints can be readily incorporated into the definition of $\mathcal{C}$, and therefore does not affect our mathematical formulation (Methods).

**AIs and mismatch classification.** While we have provided a systematic framework to probe the structure of {BS}, much insight can be gleaned from a study of AIs. AIs correspond to band insulators constructed by first specifying a symmetric set of lattice points in real space, and then fully occupying a set of orbitals on each of the lattice sites. The possible set of AIs can be easily read off from tabulated data of SGs[26, 33] (Supplementary Note 2). In addition, once the real-space degrees of freedom are specified, one can compute the corresponding element in {BS}.

As stacking two AIs lead to another AI, we see that {AI} ≤ {BS} as groups. Any subgroup of $\mathbb{Z}^{d_{BS}}$ is again a free, finitely generated Abelian group, and therefore we conclude

$$\{\text{AI}\} \simeq \mathbb{Z}^{d_{AI}} \equiv \left\{ \sum_{i=1}^{d_{AI}} m_i \mathbf{a}_i : m_i \in \mathbb{Z} \right\}, \tag{3}$$

where we denote by $\{\mathbf{a}_i\}$ a complete set of basis for {AI}.

Once {BS} and {AI} are separately computed, it is straightforward to evaluate the quotient group (Supplementary Note 3)

$$X_{BS} \equiv \frac{\{\text{BS}\}}{\{\text{AI}\}}. \tag{4}$$

Physically, an entry in $X_{BS}$ corresponds to an infinite class of BSs that, while distinct as elements of {BS}, only differ from each other by the stacking of an AI. By definition, the entire subgroup {AI} collapses into the trivial element of $X_{BS}$. Conversely, any nontrivial element of $X_{BS}$ corresponds to BSs that cannot be be interpreted as AIs, and therefore $X_{BS}$ serves as a symmetry indicator of topological BSs. One can further show that every element of $X_{BS}$ can be realized by a physical BS (Methods), and therefore $X_{BS}$ indeed corresponds to indicators of band topology in physical systems.

Following the described recipe, we compute {AI}, {BS}, and $X_{BS}$ for all 230 3D SGs in the four symmetry settings mentioned. Results for spinful fermions with TR symmetry, relevant for real materials with or without spin–orbit coupling and no magnetic order, are tabulated in Tables 1–4. The results for other symmetry settings and dimensions (Methods) are presented in Supplementary Tables 5–20.

An interesting observation from this exhaustive computation is the following: for all the symmetry settings considered, we found $d_{BS} = d_{AI}$, and therefore $X_{BS}$ is always a finite Abelian group. Equivalently, when only symmetry labels are used in the diagnosis, a BS is nontrivial precisely when it can only be understood as a fraction of an AI. In addition, $d_{BS} = d_{AI}$ implies that a complete set of basis for {BS} can be found by studying combinations of AIs, similar to Eq. 3 but with a generalization of the expansion coefficients $m_i \in \mathbb{Z}$ to $q_i \in \mathbb{Q}$, subjected to the constraint that the sum remains integer-valued. Although the full set of compatibility relations is needed in our computation establishing $d_{BS} = d_{AI}$, using our results the basis of {BS} can be readily computed directly from {AI} (Supplementary Note 3). Since {BS} can be easily found this way, we will refrain from providing a lengthy list of all the bases we found.

**Table 2 Characterization of band structures for systems with time-reversal symmetry and negligible spin–orbit coupling**

| d | Space groups |
|---|---|
| 1 | 1, 4, 7, 9, 19, 29, 33, 76, 78, 144, 145, 169, 170 |
| 2 | 8, 31, 36, 41, 43, 80, 92, 96, 110, 146, 161, 198 |
| 3 | 5, 6, 18, 20, 26, 30, 32, 34, 40, 45, 46, 61, 106, 109, 151, 152, 153, 154, 159, 160, 171, 172, 173, 178, 179, 199, 212, 213 |
| 4 | 24, 28, 37, 39, 60, 62, 77, 79, 91, 95, 102, 104, 143, 155, 157, 158, 185, 186, 196, 197, 210 |
| 5 | 3, 14, 17, 27, 42, 44, 52, 56, 57, 94, 98, 100, 101, 108, 114, 122, 150, 156, 182, 214, 220 |
| 6 | 11, 15, 35, 38, 54, 70, 73, 75, 88, 90, 103, 105, 107, 113, 142, 149, 167, 168, 184, 195, 205, 219 |
| 7 | 13, 22, 23, 59, 64, 68, 82, 86, 117, 118, 120, 130, 163, 165, 180, 181, 203, 206, 208, 209, 211, 218, 228, 230 |
| 8 | 21, 58, 63, 81, 85, 97, 116, 133, 135, 137, 148, 183, 190, 201, 217 |
| 9 | 2, 25, 48, 50, 53, 55, 72, 99, 121, 126, 138, 141, 147, 188, 207, 216, 222 |
| 10 | 12, 74, 93, 112, 119, 176, 177, 202, 204, 215 |
| 11 | 66, 84, 128, 136, 166, 227 |
| 12 | 51, 87, 89, 115, 129, 134, 162, 164, 174, 189, 193, 223, 226 |
| 13 | 16, 67, 111, 125, 194, 224 |
| 14 | 49, 140, 192, 200 |
| 15 | 10, 69, 71, 124, 127, 132, 187 |
| 17 | 225, 229 |
| 18 | 65, 83, 131, 139, 175 |
| 22 | 221 |
| 24 | 191 |
| 27 | 47, 123 |

*d* the rank of the Abelian group formed by the set of band structures

**Table 3 Symmetry-based indicators of band topology for systems with time-reversal symmetry and significant spin–orbit coupling**

| $X_{BS}$ | Space groups |
|---|---|
| $\mathbb{Z}_2$ | 81, 82, 111, 112, 113, 114, 115, 116, 117, 118, 119, 120, 121, 122, 215, 216, 217, 218, 219, 220 |
| $\mathbb{Z}_3$ | 188, 190 |
| $\mathbb{Z}_4$ | 52, 56, 58, 60, 61, 62, 70, 88, 126, 130, 133, 135, 136, 137, 138, 141, 142, 163, 165, 167, 202, 203, 205, 222, 223, 227, 228, 230 |
| $\mathbb{Z}_8$ | 128, 225, 226 |
| $\mathbb{Z}_{12}$ | 176, 192, 193, 194 |
| $\mathbb{Z}_2 \times \mathbb{Z}_4$ | 14, 15, 48, 50, 53, 54, 55, 57, 59, 63, 64, 66, 68, 71, 72, 73, 74, 84, 85, 86, 125, 129, 131, 132, 134, 147, 148, 162, 164, 166, 200, 201, 204, 206, 224 |
| $\mathbb{Z}_2 \times \mathbb{Z}_8$ | 87, 124, 139, 140, 229 |
| $\mathbb{Z}_3 \times \mathbb{Z}_3$ | 174, 187, 189 |
| $\mathbb{Z}_4 \times \mathbb{Z}_8$ | 127, 221 |
| $\mathbb{Z}_6 \times \mathbb{Z}_{12}$ | 175, 191 |
| $\mathbb{Z}_2 \times \mathbb{Z}_2 \times \mathbb{Z}_4$ | 11, 12, 13, 49, 51, 65, 67, 69 |
| $\mathbb{Z}_2 \times \mathbb{Z}_4 \times \mathbb{Z}_8$ | 83, 123 |
| $\mathbb{Z}_2 \times \mathbb{Z}_2 \times \mathbb{Z}_2 \times \mathbb{Z}_4$ | 2, 10, 47 |

$X_{BS}$ the quotient group between the group of band structures and that of atomic insulators

To illustrate the ideas more concretely, we discuss a simple example concerning non-TR-symmetric spinless fermions symmetric under SG 106. In this setting, $d_{BS} = d_{AI} = 3$, and $\mathbf{a}_1$, one of the three generators of {AI}, has the property that all irreps appear an even number of times, while the other two generators contain some odd entries. Now consider $\mathbf{b}_1 \equiv \mathbf{a}_1/2$, which is still integer-valued. Clearly, by linearity $\mathbf{b}_1$ satisfies all symmetry constraints, and therefore $\mathbf{b}_1 \in$ {BS}. However, $\mathbf{b}_1 \notin$ {AI} $\equiv \left\{ \sum_{i=1}^{3} m_i \mathbf{a}_i : m_i \in \mathbb{Z} \right\}$, and therefore $\mathbf{b}_1$ corresponds to a quantum BS, and indeed it is a representative for the nontrivial element of $X_{BS} = \mathbb{Z}_2$. In addition, if we consider a tight-binding model with a representation content corresponding to $\mathbf{a}_1$, the decomposition $\mathbf{a}_1 = \mathbf{b}_1 + \mathbf{b}_1$ implies that it is possible to open a band gap at all high-symmetry momenta at half filling, and thereby realizing the quantum BS $\mathbf{b}_1$. It turns out that, in fact, $\mathbf{b}_1$ corresponds to a filling-enforced QBI (feQBI)[30]. We will elaborate further on this point in the Supplementary Note 4.

Before we move on to concrete applications of our results, we pause to clarify some subtleties in the exposition. Recall that the notion of BS is defined using the presence of band gaps at all high-symmetry momenta. Generally, however, there can be gaplessness in the interior of the Brillouin zone that coexist with our definition of BS. While in some cases such gaplessness is accidental in nature, in the sense that it can be annihilated without affecting the BS, in some more interesting cases it is enforced by the specification of the symmetry content. This was pointed out in refs [13, 14] for inversion-symmetric systems without TR symmetry, where certain assignments of the parity eigenvalues ensure the presence of Weyl points at some generic momenta. When a nontrivial element in $X_{BS}$ can be insulating, we refer to it as as a reQBI; when it is necessarily gapless, we call it a representation-enforced semimetal (reSM). We caution that $X_{BS}$ will naturally include both reQBIs and reSMs, although

some symmetry settings naturally forbid the notion of reSMs. In fact, one can show that their individual diagnoses are related by $X_{SM} = X_{BS}/X_{BI}$ (Supplementary Note 5). Hence, given an entry of $X_{BS}$ one has to further decide whether it corresponds to a reSM or a reQBI. In Supplementary Note 5, we provide general arguments on the existence of reSMs for systems with significant spin–orbit coupling.

In addition, we also note that, while every BS belonging to a nontrivial class of $X_{BS}$ is necessarily nontrivial, some systems in the trivial class can also be topological. By definition, the representation content of a BS belonging to the trivial class of $X_{BS}$ can be constructed by stacking of AIs. However, if the stacking necessarily involves negative coefficients, the BS cannot be attained from stacking physical AIs, and therefore is still topologically nontrivial. Some of the feQBIs discussed in ref. [30] also fall into this category. Alternatively, when the topological nature of a system is undetectable using only symmetry labels, say for the tenfold-way phases in the absence of any spatial symmetries beyond the lattice translations, the system belongs to the trivial element of $X_{BS}$ despite it is topological. The general relation between $X_{BS}$ and the conventional tenfold-way classification depends on the symmetry setting at hand, and its understanding is an important open question (Methods).

**Quantum Band Insulators in conventional settings**. Having derived a general theory for finding symmetry-based indicators of band topology, we now turn to applications of the results. As a first application, we utilize the results in Table 3 to look for reQBIs that are not diagnosed by previously available topological invariants. In particular, we will focus on a result concerning one of the most well-studied symmetry setting: materials with significant spin–orbit coupling symmetric under TR, lattice translations and inversion (SG 2).

As shown in Table 3, $X_{BS} = (\mathbb{Z}_2)^3 \times \mathbb{Z}_4$ for SG 2. Using the Fu-Kane criterion[8], one can verify that the strong and weak TIs, respectively, serve as the generators of the $\mathbb{Z}_4$ and $\mathbb{Z}_2$ factors. This identification, however, fails to account for the nontrivial nature of the doubled strong TI, which being a nontrivial element in $\mathbb{Z}_4$ corresponds to a reQBI. It is also not covered in the earlier lines

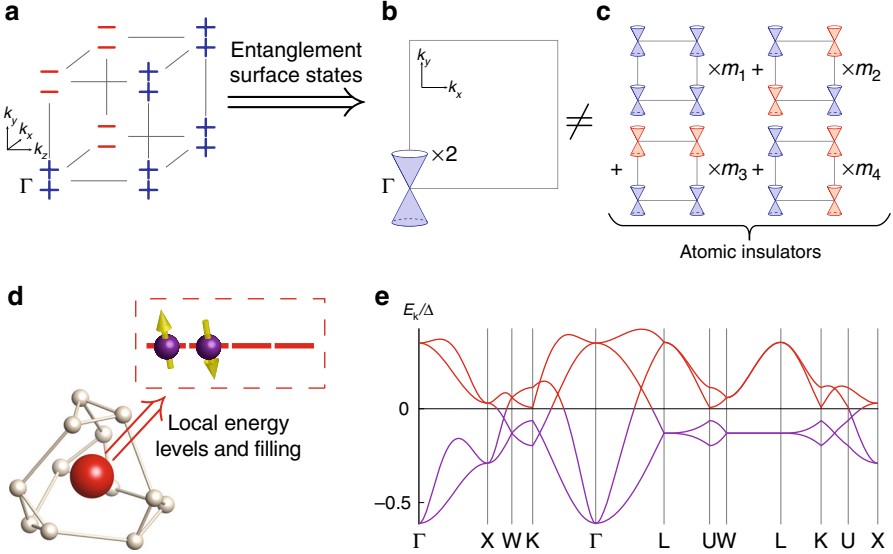

**Fig. 2** Examples of topological band structures. **a–c** A representation-enforced quantum band insulator of spinful electrons with time-reversal and inversion symmetries, dubbed the "doubled strong TI". **a** Using the Fu-Kane parity criterion[8], the strong and weak $\mathbb{Z}_2$ indices can be computed from the the parities of the occupied bands, which we indicate by $\pm$ at the eight time-reversal invariant momenta. Shown are the parities of one state from each Kramers pair for a doubled strong TI with four *filled bands*. **b** The entanglement spectrum at a spatial cut, parallel to the $x$–$y$ plane and containing an inversion center, features two Dirac cones at $\Gamma$[13, 14, 16]. Such Dirac cones are known to possess integer-valued charges under the inversion symmetry, and we denote the positively charged and negatively charged cones, respectively, by *blue* and *red*. **c** Inversion-symmetric atomic insulators feature entanglement surface Dirac cones in general, but their presence depends on the arbitrary choice of the cut. We find that the possible Dirac-cone arrangement arising from atomic insulators can only be a linear combination of four basic configurations, illustrated as a sum with the integral weights $m_i$. The arrangement in **b** cannot be reconciled with those in **c**, confirming the nontriviality of the doubled strong TI. **d, e** Example of a lattice-enforced semimetal for spinful electrons with time-reversal symmetry. **d** We consider a site (*red sphere*) under a local environment (*beige*) symmetric under the point group $T$, and suppose the relevant local energy levels form the four-dimensional irreducible representation, which is half-filled (*boxed*). **e** When the *red site* sits at the highest-symmetry position of space group 219, the specified local energy levels and filling gives rise to a half-filled eight-band model (each band shown is doubly degenerate). Such (semi-)metallic behavior is dictated by the specification of the microscopic degrees of freedom in this model

of work focusing on inversion-symmetric insulators[13, 14, 16]. This reQBI possesses a trivial magnetoelectric response ($\theta = 0$) and is not expected to have protected surface states.

Nonetheless, the nontrivial nature of the reQBI can be seen from its entanglement spectrum, which exhibits protected gaplessness related to the parity eigenvalues of the filled bands (Fig. 2a). In the present context, we define the entanglement spectrum as the collection of single-particle entanglement energies arising from a spatial cut, which contains an inversion center and is perpendicular to a crystalline axis. Refs [13, 14, 16] showed that the entanglement spectrum of TR and inversion symmetric insulators generally have protected Dirac cones at the TR invariant momenta of the surface Brillouin zone. These Dirac cones carry effective integer charges under inversion symmetry, and as a result they are symmetry-protected. The doubled strong TI phase has twice the number of Dirac cones as the regular strong TI (Fig. 2b).

Yet, one must use caution in interpreting the nontrivial nature of such entanglement, since inversion-symmetric AIs also have protected entanglement surface states whenever the center of mass of an electronic wavefunction is pinned to the entanglement cut. The presence of these entanglement signatures, however, is dependent on the arbitrary choice of the location of the cut, and therefore is not as robust as the other topological characterizations. In contrast, since we have already quotient out all AIs in the definition of $X_{BS}$, the reQBI at hand must have a more topological origin. This is verified from the pictorial argument in Fig. 2a–c, where we contrast the entanglement spectrum of the doubled strong TI with those that can arise from AIs. Importantly, we see that the total Dirac-cone charge of an AI is always 0 mod 4, whereas the doubled strong TI has a charge of 2 mod 4. This

implies that the entanglement gaplessness is independent of the arbitrary choice of the cut, and in fact shows that the bulk computation of $X_{BS}$ can be reproduced by considering the entanglement spectrum for this symmetry setting. Note that, if TR is broken, Kramers paring will be lifted and the irrep content of this reQBI becomes achievable with an AI. This suggests that the reQBI at hand is protected by the combination of TR and inversion symmetry. It is an interesting open question to study whether or not this reQBI has any associated quantized physical response[13].

We note that, since the strong TI is compatible with any additional spatial symmetry, the argument above is applicable to any centrosymmetric SGs. Indeed, as can be seen from Table 3, all of them have $|X_{BS}| \geq 4$, consistent with our claim. Therefore, the doubled strong TI phase could be realizable in a large number of materials classes. Finally, we remark that the same $X_{BS}$ is found for SG 2 in all the other symmetry settings, although their physical interpretations are different. In particular, the generators of $\mathbb{Z}_4 < X_{BS}$ correspond to a reSM in the other settings. This observation also shows that the doubled strong TI phase remains nontrivial in the absence of spin–orbit coupling.

**Lattice-enforced semimetals.** As another application of our results, we demonstrate how the structure of {BS} exposes constraints on the possible phases of a system arising from the specification of the microscopic degrees. We will in particular focus on the study of semimetals, but a similar analysis can be performed in the study of, say, reQBIs.

As a warm-up, recall the physics of (spinless) graphene, where specifying the honeycomb lattice dictates that the irrep at the K

**Table 4 Symmetry-based indicators of band topology for systems with time-reversal symmetry and negligible spin–orbit coupling**

| $X_{BS}$ | Space groups |
|---|---|
| $\mathbb{Z}_2$ | 3, 11, 14, 27, 37, 48, 49, 50, 52, 53, 54, 56, 58, 60, 66, 68, 70, 75, 77, 82, 85, 86, 88, 103, 124, 128, 130, 162, 163, 164, 165, 166, 167, 168, 171, 172, 176, 184, 192, 201, 203 |
| $\mathbb{Z}_2 \times \mathbb{Z}_2$ | 12, 13, 15, 81, 84, 87 |
| $\mathbb{Z}_2 \times \mathbb{Z}_4$ | 147, 148 |
| $\mathbb{Z}_2 \times \mathbb{Z}_2 \times \mathbb{Z}_2$ | 10, 83, 175 |
| $\mathbb{Z}_2 \times \mathbb{Z}_2 \times \mathbb{Z}_2 \times \mathbb{Z}_4$ | 2 |

$X_{BS}$ the quotient group between the group of band structures and that of atomic insulators

point is necessarily 2D, and therefore the system is guaranteed to be gapless at half filling. Using the structure of {BS} we described, this line of reasoning can be efficiently generalized to an arbitrary symmetry setting: any specification of the lattice degrees of freedom corresponds to an element $\mathbf{A} \in$ {AI}, and one simply asks if it is possible to write $\mathbf{A} = \mathbf{B}_\nu + \mathbf{B}_c$, where $\mathbf{B}_{\nu,c} \in$ {BS} satisfies the physical non-negative condition, such that $\mathbf{B}_\nu$ corresponds to a BS with a specified filling $\nu$. Whenever the answer is no, the system is guaranteed to be (semi-)metallic. We refer to any such system as a leSM. Note that a stronger form of symmetry-enforced gaplessness can originate simply from the electron filling, and such systems were dubbed as filling-enforced semimetal (feSMs)[34, 35]. We will exclude feSMs from the definition of leSMs, i.e., we only call a system a leSM if the filling $\nu$ is compatible with some band insulators in the same symmetry setting, but is nonetheless gapless because of the additional lattice constraints.

A preliminary analysis reveals that leSMs abound, especially for spinless systems with TR symmetry. This is in fact anticipated from the earlier discussions in refs [36–38]. Instead, we will turn our attention to TR-invariant systems with significant spin–orbit coupling, which lies beyond the scope of these earlier studies and oftentimes leads to interesting physics[1, 7, 30]. A systematic survey of them will be the focus of another study. Here, we present a proof-of-concept leSM example we found, which arises in systems symmetric under SG 219 ($F\bar{4}3c$). We will only sketch the key features of the model, and the interested readers are referred to the Methods section for details of the analysis.

We consider a lattice with two sites in each primitive unit cell, and that each site has a local environment corresponding to the cubic point group $T$ (Fig. 2d). We suppose the relevant on-site degrees of freedom transform under the four-dimensional irreducible co-representation of $T$ under TR symmetry[25], and that the system is at half filling, i.e., the filling is $\nu = 4$ electrons per primitive unit cell. Although the local orbitals are partially filled, generically a band gap becomes permissible once electron hopping is incorporated. Naively, for the present problem this may appear to be the likely scenario, since the momentum-space irreps all have dimensions $\leq 4$[25] and band insulators are known to be possible at this filling[34, 35]. However, using our framework one can prove that no BS is possible for this system at $\nu = 4$, implying that there is irremovable lattice-enforced gaplessness at some high-symmetry line. This is indeed verified in Fig. 2e, where we plot the BS obtained from an example tight-binding model (Methods).

## Discussion

In this work, we present a simple mathematical framework for efficiently analyzing BSs as entities defined globally over the Brillouin zone. We further utilize this result to systematically quantify the mismatch between the momentum-space and real-space descriptions of free electron phases, obtaining a plethora of symmetry settings for which topological materials are possible.

Our results concern a fundamental aspect of the ubiquitous band theory. For electronic problems, we demonstrated the power of our approach by discussing three particular applications, predicting both QBIs and semimetals (see also Supplementary Note 4). We highlight four interesting future directions below: first, to incorporate the tenfold-way classification into our symmetry-based diagnosis of topological materials[28]; second, to discover quantized physical responses unique to the phases we predicted[13, 14, 18]; third, to extend the results to magnetic SGs[25]; and lastly, to screen materials database for topological materials relying on fast diagnosis invoking only symmetry labels[39]. More broadly, we expect our analysis to shed light on any other fields of studies, most notably photonics and phononics, where the interplay between topology, symmetry, and BSs is of interest.

Note added: Recently, ref. [40] appeared, which has some overlap with the present work, in that it also identifies topological band insulators by contrasting them with AIs. However, the present work differs from ref. [40] in important ways in the formulation of the problem and the mathematical approach adopted.

## Methods

**Glossary of abbreviations.** For brevity, we have introduced several abbreviations in the text. For the readers' convenience, we provide a glossary of the less-standard ones here.

AI (atomic insulator): band insulators possessing localized symmetric Wannier functions.

BS (band structure): a set of energy bands separated from all others by band gaps above and below at all high-symmetry momenta.

fe (filling-enforced): referring to attributes that follow from the electron filling of the system.

le (lattice-enforced): referring to attributes that follow from the specification of the microscopic degrees of freedom in the lattice.

QBI (quantum band insulators): band insulators, with or without protected surface states, that do not admit any atomic limit provided the protecting symmetries are preserved.

re (representation-enforced): referring to attributes that follow from knowledge on the symmetry representations of the energy bands.

SG (space group): any one of the 230 spatial symmetry groups of crystals in three dimensions.

SM (semimetals): filled bands with gap closings that are stable to infinitesimal perturbations.

TI (topological insulator): $\mathbb{Z}_2$ TIs in two or three dimensions for spin–orbit-coupled system with TR symmetry (note that we use this phrase in a restricted sense in this work).

**Three-dimensional BSs.** In the main text, we have illustrated the definition and interpretation of {BS} using a simple 1D example. In the following we summarize the key generalizations required to address 3D systems. A more detailed discussion is presented in Supplementary Notes 1 and 2.

Similar to the 1D example, in the general 3D setting a collection of integers, corresponding to the multiplicities of the irreps in the BS, is assigned to each high-symmetry momentum. By the gap condition imposed in the definition of a BS, these integers are invariant along high-symmetry lines. In addition, any pair of symmetry-related momenta will share the same labels. Altogether, we see that the symmetry content of a BS, together with the number of bands $\nu$, is similarly specified by a finite number of integers, which we call $D$. Therefore, these symmetry labels can be identified as elements of the group $\mathbb{Z}^D$, where group addition corresponds to the physical operation of stacking BSs. As discussed, however, these integers are again subjected to the compatibility relations, which arise whenever a high-symmetry momentum is continuously connected to another with a lower symmetry. By continuity, the symmetry content of the BS at the lower-symmetry momentum is fully specified by that of the higher-symmetry one, giving rise to linear constraints we denote collectively by the matrix $\mathcal{C}$. The group {BS} is then defined as in Eq. 1, and again we find

$$\{BS\} \equiv \ker \mathcal{C} \cap \mathbb{Z}^D \simeq \mathbb{Z}^{d_{BS}}, \tag{5}$$

where as before $d_{BS} = \dim \ker \mathcal{C}$. Note that this result has a simple geometric interpretation: from the definition Eq. 1, we can picture $\ker \mathcal{C}$ as a $d_{BS}$-dimensional hyperplane slicing through the hypercubic lattice $\mathbb{Z}^D$ embedded in $\mathbb{R}^D$ (Supplementary Note 3). This gives rise to the sublattice $\mathbb{Z}^{d_{BS}}$ (Fig. 1b).

**Effect of TR symmetry.** Being anti-unitary, TR alone does not modify the irreps. However, under the action of TR an irrep can be paired to either a distinct copy of itself, or to another irrep[25]. The constraints arising from both cases can be readily incorporated into the definition of $\mathcal{C}$: when under TR an irrep $\alpha$ at $\mathbf{k}$ is paired with a different irrep $\beta$ at $\mathbf{k}'$, where $\mathbf{k}' = \mathbf{k}$ or $\mathbf{k}' = -\mathbf{k}$, we simply add to $\mathcal{C}$ an additional compatibility relation $n_{\mathbf{k}}^{\alpha} = n_{\mathbf{k}'}^{\beta}$; when $\alpha$ is paired with itself, we demand $\alpha$ to be an even integer, which can be achieved by redefining $\tilde{n}_{\mathbf{k}}^{\alpha} \equiv n_{\mathbf{k}}^{\alpha}/2$ and a corresponding rewriting of $C$ in terms of $\tilde{\mathbf{n}}$.

Note that, although TR is not included in our definition of high-symmetry momenta, we will always take Kramers degeneracy in spin–orbit-coupled systems into account.

**Physical aspects of the mathematical treatment.** While we have shown in the main text that {BS} is a well-defined mathematical entity and identified its general structure, it remains to connect it to the study of physical BSs. Here, we first argue that as long as $\mathbf{B} \in \{BS\}$ satisfies the physical condition of non-negativity, namely all entries of $\mathbf{B}$ are non-negative integers, then $\mathbf{B}$ corresponds to a physically realizable BS. Next, we will show that all entries of $X_{BS}$ have physical representatives. Below, we will only sketch the arguments involved. Interested readers are referred to Supplementary Note 3 for a more elaborated discussion.

Recall that in motivating the definition Eq. 1, in order to obtain a group structure we have lifted the physical condition that all irreps must appear a non-negative number of times. This implies that any physical BS must correspond to elements in the subset $\{BS\}_P = \ker \mathcal{C} \cap \mathbb{Z}_{\geq 0}^D \subset \{BS\}$. However, one should question whether all elements in $\{BS\}_P$ indeed correspond to some physical BSs. This can be reasoned by noting that as {BS} is defined as the solution of all compatibility relations, all necessary band crossings and degeneracies have been taken into account. Therefore, by adjusting the energetics of a sufficiently general physical model one can realize any element of $\{BS\}_P$, up to accidental degeneracies that can be removed by symmetry-preserving perturbations.

Next, we argue that all entries in $X_{BS}$ have physical representatives. Suppose an element of $X_{BS}$ is represented by a $\mathbf{B} \in \{BS\}$, which does not satisfy the physical condition of non-negativity. Using a small technical corollary we discuss in the Supplementary Note 2, one can show that the representation content of any $\mathbf{B}$ can be rectified by stacking with some $\mathbf{A} \in \{AI\}$, i.e., $\mathbf{B} + \mathbf{A} \in \{BS\}_P$. Since $\mathbf{B} + \mathbf{A}$ belongs to the same class as B in $X_{BS}$, we arrive at a physical representative of the same element of $X_{BS}$.

**Extension to other symmetry settings.** Results for TR-symmetric systems in any of the 230 SGs are presented in the main text, and the corresponding ones for systems without TR symmetry are presented in Supplementary Tables 5–8. Here, we remark that the corresponding results for quasi-1D and 2D systems, described, respectively, by rod and layer group symmetries[26, 41], can be readily obtained (Supplementary Note 3). The results are presented in Supplementary Tables 9–20. In particular, we found $X_{BS} = \mathbb{Z}_1$, the trivial group, for all quasi-1D systems. This is consistent with the picture that topological BSs in 1D can be understood as frozen polarization states, which are AIs and hence trivial in our definition.

**Relation to K-theory-based classifications.** As discussed, band topology identified within the K-theory framework may not be detectable using only symmetry labels. As an example, consider a 2D system with only lattice translation symmetries. For such systems, the K-theory classification of band insulators in refs [22, 24, 28] gives $\mathbb{Z}^2$, where the two factors correspond, respectively, to the electron filling (i.e., number of bands) and the Chern number. In contrast, within our approach we find {BS} = {AI} = $\mathbb{Z}$, since in this setting the only symmetry label is the filling, which cannot detect the Chern number of the bands. Furthermore, as there exists an AI for any filling $\nu$, we find $X_{BS} = \mathbb{Z}_1$, the trivial group.

However, in some other cases using symmetry labels alone one can also detect the tenfold-way phases, as in cases where the Fu-Kane parity criterion applies[8]. As a related problem, one can readily study how a centrosymmetric SG constrains the possible weak TI phases using our results in Table 3. This is related to the number of factors in $X_{BS}$, i.e., the number of independent generators $N_g$. As one such factor is reserved for the strong TI, the SG is compatible with at most $N_g - 1$ independent weak TI phases. While this has been pointed out for certain cases in the literature[42], our approach automatically encapsulates some of these result in a simple manner.

**Example of leSMs.** Here, we provide details on the leSM example discussed in the main text. We consider a TR-symmetric system in SG 219 with significant spin–orbit coupling. We will establish that for a particular lattice specification, a semimetallic behavior is unavoidable at a filling $\nu = 4$, although band insulators are generally possible at this filling for the present symmetry setting[34, 35]. This arises from the fact that, given the available symmetry irreps specified by the lattice, corresponding to an element $\mathbf{A} \in \{AI\}$, there is no way to satisfy all the compatibility relations at the filling $\nu = 4$, i.e., $\mathbf{A} \neq \mathbf{B}_v + \mathbf{B}_c$ for any non-zero $\mathbf{B}_v$, $\mathbf{B}_c \in \{BS\}$ satisfying the physical condition of non-negativity.

We consider a lattice in Wyckoff position $a$, which contains two sites at $\mathbf{r}_1 \equiv (0, 0, 0)$ and $\mathbf{r}_2 \equiv (1/2, 0, 0)$ in the unit cell. The two sites are related by a glide symmetry, and the site-symmetry group for each site is given by the point group $T$

(i.e., the orientation-preserving symmetries of a tetrahedron, also known as the chiral tetrahedral symmetry group). We suppose the physically relevant degrees of freedom arise from the three $p_{x,y,z}$ orbitals on each site, which together with electron spin leads to a six-dimensional local Hilbert space. We will let $\mathbf{L}$ and $\mathbf{S}$, respectively, denote the orbital and spin angular momentum operators in the single-particle basis.

As described in the main text, we consider a TR-symmetric system with a strong crystal-field splitting:

$$H_\Delta = \Delta \sum_{\mathbf{r}:\text{allsites}} \mathbf{c}_{\mathbf{r}}^{\dagger} (\mathbf{L} \cdot \mathbf{S}) \mathbf{c}_{\mathbf{r}}, \tag{6}$$

where $\mathbf{c}_{\mathbf{r}}$ represents the six-dimensional (column) vector corresponding to the internal degrees of freedom. One can verify that when $\Delta > 0$, $H_\Delta$ splits the local energy levels to a total spin-1/2 doublet lying below the total spin-3/2 multiplet. While we have chosen $H_\Delta$ to conserve the total spin $\mathbf{L} + \mathbf{S}$ for convenience, such conservation is not required by the local symmetry, which is described by the point group $T < SO(3)$. Therefore, the total spin quantum numbers are not a priori good quantum numbers for the problem at hand. However, one can verify that the spinful, TR symmetric irreps of $T$ coincide with the total spin decomposition described above[25], and hence insofar as symmetries are concerned $H_\Delta$ is a sufficiently generic crystal-field Hamiltonian. We also note that, if TR symmetry is broken, the fourfold degenerate states originating from the total spin-3/2 states can be further split.

As discussed in the main text, we are interested in the systems arising from half filling the fourfold degenerate local energy levels. To this end, we assume $\Delta$ is the dominant energy scale in the problem, which implies the low-lying doubly degenerate states can be decoupled from the description of the system as long as they are fully filled. This leaves behind the fourfold degenerate energy levels, which we assume are half-filled. As there are two symmetry-related sites in each unit cell, these considerations altogether imply that the BS around the Fermi energy is described by an effective eight-band tight-binding model at filling $\nu = 4$.

Next, we consider a nearest-neighbor hopping term

$$H_{t,\lambda} = \sum_{g \in \mathcal{G}} g \Big( \mathbf{c}_{\mathbf{r}_1}^{\dagger} (t + \lambda \, \hat{\mathbf{x}} \cdot (\mathbf{L} \times \mathbf{S})) \mathbf{c}_{\mathbf{r}_2} \Big) g^{\dagger} + \text{h.c.}, \tag{7}$$

where h.c. denotes Hermitian conjugate, and the notation $\sum_{g \in \mathcal{G}} g(\dots) g^{\dagger}$ denotes all the terms generated by transforming the terms in the parenthesis by the symmetry elements of the SG $\mathcal{G}$.

The BS of the full Hamiltonian $H = H_\Delta + H_{t,\lambda}$ is shown in Fig. 2e, with parameters $(t/\Delta, \lambda/\Delta) = (0.01, 0.05)$. Note that we have only shown the eight bands near the Fermi energy; four fully filled bands arising from the doubly degenerate local orbitals are separated in energy by $\mathcal{O}(\Delta)$. As our computation dictates, the lattice specification gives rise to energy bands that are necessarily gapless along the high-symmetry lines at filling $\nu = 4$. Interestingly, note that the lattice-enforced gaplessness is of a more subtle flavor: unlike spinless graphene, where the the gaplessness is enforced by the dimensions of the irreps involved, here all the irreps have dimensions $\leq 4$, and therefore the impossibility of finding a BS at $\nu = 4$ is reflected in the connectivity of the energy bands.

In closing, we remark that the notion of leSM is not as robust as the other notions we introduced in this work, say feSM or reQBI. Specifically, the (semi-)metallic behavior of the system is protected by the specification of the microscopic degrees of freedom, which is only sensible assuming a certain knowledge about the energetics of the problem. Under stacking of a trivial phase, say when we incorporate into the description a set of fully filled bands corresponding to an AI, the enforced gaplessness may become unstable, as these apparently inert degrees of freedom can also supply the representations needed to open a gap at the targeted filling. This can be readily seen from the example above: if we switch the sign of $\Delta$, the same electron filling will now correspond to the full filling of the fourfold degenerate multiplet on each site, which leads to an AI. Such instability should be contrasted with, say, the notion of reQBIs, which by definition remains nontrivial as long as the extra degrees of freedom we introduce are in the trivial class, i.e., correspond to AIs.

**Data availability.** All relevant data are available from the authors upon reasonable request.

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

## Acknowledgements

We thank C.-M. Jian, A. Turner and M. Zaletel for insightful discussions and collaborations on earlier works. We also thank C. Fang for useful discussions. A.V. and H.C.P. were supported by NSF DMR-1411343. A.V. acknowledges support from a Simons Investigator Award. H.W. acknowledges support from JSPS KAKENHI Grant Number JP17K17678.

## Author contributions

All authors contributed to all aspects of this work.

## Additional information

**Competing interests:** The authors declare no competing financial interests.

**Change History:** A correction to this article has been published and is linked from the HTML version of this article.

