## [Peer Review File · Nature Communications]

REVIEWERS' COMMENTS:

Reviewer #1 (Remarks to the Author):

Dear Editor

I have reviewed the work of Hoi Chun Po et al and I found it very interesting and worthy of publication in Nature Communications. Although the whole work is actually quite long, due to lengthy Supplementary Material, the description of the methodologies and the conclusions can be effectively communicated within the main text. These ideas will be easily picked up by professional crystallographers but other condensed matter scientists might need to dust off their textbooks on representations of space groups, such as Ref. [20] (by any means, this will do good for anybody).

The work under review builds on the observation from Refs. [13,14] that both the topologically trivial and non-trivial representations of a "space-group" which includes only inversion can be classified by the eigenvalues of the inversion operator at the high-symmetry points. The present work is, in some sense, a generalization of that observation to all 230 space groups in 3-dimensions. More concretely, the authors define invariants and numerical invariants in terms of a space of representation-coefficients and the dimension of this space, respectively. These are computed explicitly for all 230 space groups. The representations coming from atomic limits are factor out, hence these spaces contain information about the topologically non-trivial band-structures which cannot be dis-entangled in independent flat bands without closing the gap or breaking the space-group symmetry.

I have a few comments, which I am sure the authors can be easily address:

1)In this reviewer's opinion, the "large zoo of irreps" cannot be entirely left the Appendix, and more substance needs to be added to Eq. 2.

2)Even in the appendix where the irreps are discussed, the presentation is not so clear. For example, the summary in Ref. [20] at pages 151-152 (in the 2009 print of the book) on how ALL irreducible reps of a space group is constructed should be included in one form or another.

3)Some of the conclusions in D2 about the physical content are important and should be presented in the main text.

4)Since the BS is defined by a gap condition over the highly-symmetric points of the Brillouin zone (only), the present analysis allows for gap-closings at regular k-points. As such, the topological phases identified this way can correspond to either insulator or semi-metals. Can these two cases be differentiated by arguments similar to those related to Eq. A3? More precisely, for the metallic case one should find that the representations cannot be extended continuously over the whole Brillouin zone (hence they must be connected to other bands).

5)It is unfortunate that some existing and highly regarded works from mathematical physics literature have been not cited. Some of these works are:

Ken Shiozaki, Masatoshi Sato, Kiyonori Gomi, "Topological Crystalline Materials - General Formulation, Module Structure, and Wallpaper Groups" arXiv:1701.08725 (2017)

Kiyonori Gomi, "Twists on the Torus Equivariant under the 2-Dimensional Crystallographic Point Groups", SIGMA 13, 014 (2017).

Ken Shiozaki, Masatoshi Sato, Kiyonori Gomi, "Z2 topology in nonsymmorphic crystalline insulators: Möbius twist in surface states", PRB 91, 155120 (2015).

Ken Shiozaki, Masatoshi Sato, Kiyonori Gomi, "Topology of nonsymmorphic crystalline insulators and superconductors", PRB 93, 195413 (2016)

Reviewer #2 (Remarks to the Author):

Review on "Complete Theory of Symmetry-based Indicators of Band Topology" by Hoi Chun Po, Ashvin Vishwanath and Haruki Watanabe.

I have read the manuscript as well as the refs [13] and [23] in the paper with care. In essence the subject of new classification procedures and associated physical consequences in presence of space group symmetries is rather timely. Not in the least place due to the recent work, refs [13] and [23], and work by the Princeton group.

The authors exploit a recent connection between K-theory and combinatorial constraints from ref [13] and their previous work on the essential role of the filling constraint, ref [23]. All in all, this means that the bands have to be glued together in a symmetry respecting fashion, connecting high symmetry points in a certain number of possible manners. Taking into account filling constraints one can determine the system and also point out the different "topological possibilities", in the view of the paper. The extended procedure by the authors then conspires in a somewhat related framework. That is, the band structure is essentially determined by the kernel of the matrix of constraints [after lifting it to integers instead of non negative integers], Eq. 1. This structure thus reveals a natural abelian group, their $Z^{\{d_{BS}\}}$. Most importantly, the authors then extend and contrast these insights with the Atomic, that is trivial, insulator (AI). This reveals that the topological variants can be seen as a "fraction" of the AIs and results in Eq. 5.

Apart from technicalities this essentially pertains to rather heuristic insights. Indeed physically it boils down to the fact that for an AI one can find a localized states, the Wannier states, whereas for the topological counter parts this had been long known not to be possible. These "quantum band insulators" (QBIs) arising by the mismatch thus, in very crude essence, form the class of other states, such as topological insulators and relate to the integers/constraints as described. With these insights in hand the authors also demonstrate new insulating cases and semimetallic cases falling within this paradigm.

I find the paper interesting and the presentation very clear apart from some minor issues. The question is thus whether the content meets the standards of Nature Communications. The framework is interesting, fairly general and clear, but also very much an extension of the insights of the previous refs indicated. Nonetheless the authors also propose a new quantum band insulator and the lattice-enforced semimetal. Problem though is that, focussing on the first one for example, the physical effects are not clear; the new quantum band insulator has no protected surface states and a trivial magnetoelectric response. From a some viewpoints one might thus question whether we are not dealing with a nomenclature for band theory. On the other hand, from the entanglement spectrum one does see a difference.

Taking these thoughts into account I am inclined towards endorsing publication in combination with the incorporation of some suggestions.

-As said the new phases are the highlight of the paper. These have therefore to be explained a bit better in my opinion. I understand the motivation, but the crucial difference, the entanglement spectrum of the 'doubled' strong TI should then be shown in the figure.

-More generally the figures are essential part of the paper, in the text it is often referred to as an explanation. Also many readers may focus on the figures. Therefore the captions have to be improved a lot. They should be self-contained. Eg. in fig 1, the a and b panel can be linked more

to explain better that the ν and constraints are linked in a particular fashion. Maybe the authors want to interchange here the a and b panel. Also in figs 2 b and c the left part is unclear. I get what they are saying from the text, but in the figure itself it is rather obscure what the authors like to convey. For 2c one just sees an insulating band structure. It is known that these can be different from the AIs. So explain this with a understandable cartoon, I presume the left part, or leave it out.

-Also the captions of the tables should be expanded. Readers skimming through the text should be guided what the tables convey.

-On page 6 the authors consider the spin quantum numbers $1/2 \oplus 3/2$ these are for J. So please name it to make it more clear. Again either attach all the labels or just name the multiplets in words.

-Having read ref 23, I think the authors should change
" For instance, in Ref. [23], which focuses
on systems in the 17 2D wallpaper groups without any additional
symmetry, such an approach was adopted to help
develop a more physical understanding of the mathematical
treatment of Ref. [24]"

to

" For instance, in Ref. [23], which focuses
on systems in the wallpaper groups without any additional
symmetry, such an approach was adopted to help
develop a more physical understanding of the mathematical
treatment of Ref. [24]"

on page 1. The method is focussed indeed on 2d but in fact general.

-The authors should remove complete from the title. The precise connection to the mathematical structure is not yet completely identified. Also things like half mirror Chern numbers are not always picked up by these relations and should therefore maybe be re-evaluated.

Reviewer #3 (Remarks to the Author):

This manuscript presents a mathematical framework allowing to identify topological band structures in solids. More precisely its ambition is to define a theory which allows to explore all space groups of solids and identify topological properties tightly bound to these crystalline symmetries. The tools and the method for this task are defined in this paper, while the exhaustive search is postponed to future work. As such, this manuscript is naturally rather abstract and focuses more on methodology than on physical result. From this perspective, while presenting new material, it is related to several recent papers, such as references [24] and [23]. Overall, I have found this manuscript rather clear and pedagogical. The core of this paper corresponds to a discussion of the method, while all technical details are postponed to the supplementary materials (and this one is very dense). However, in a paper which already requires some special attention due to its technical nature, the frequent use of acronyms really hampers the reading. The final list is rather long (TR, SOC, QBI, BS, AI, feQBI, reQBI, reSM, DOF, leSM, feSM,...). Notwithstanding this annoyance, the reading of this manuscript was pleasant and very interesting. In this end, my opinion is that this manuscript is a very formal one. However its subject and results are of high importance in the recent domain of topological phases. The experimental search for topological phases has strongly relied on formulation of topological invariants restricted to the

presence of special symmetries such as inversion, or crystalline symmetries for semi-metals. Nevertheless no general framework starting from these symmetries existed prior to this paper. It is my understanding that such a framework will be of very high relevance for the search of new topological materials of various nature, and that this work should be published in Nature Communications.

Finally let, me had a minor comment concerning the "applications" section of this article, which I find not illustrative enough. In particular, I wonder whether part of the discussion of sec. VI of the supplementary should be included on the main text.

Reviewer #1

Comment 1.1

I have reviewed the work of Hoi Chun Po et al and I found it very interesting and worthy of publication in Nature Communications. Although the whole work is actually quite long, due to lengthy Supplementary Material, the description of the methodologies and the conclusions can be effectively communicated within the main text. These ideas will be easily picked up by professional crystallographers but other condensed matter scientists might need to dust off their textbooks on representations of space groups, such as Ref. [20] (by any means, this will do good for anybody).

The work under review builds on the observation from Refs. [13,14] that both the topologically trivial and non-trivial representations of a “space-group” which includes only inversion can be classified by the eigenvalues of the inversion operator at the high-symmetry points. The present work is, in some sense, a generalization of that observation to all 230 space groups in 3-dimensions. More concretely, the authors define invariants and numerical invariants in terms of a space of representation-coefficients and the dimension of this space, respectively. These are computed explicitly for all 230 space groups. The representations coming from atomic limits are factor out, hence these spaces contain information about the topologically non-trivial band-structures which cannot be dis-entangled in independent flat bands without closing the gap or breaking the space-group symmetry.

Reply 1.1 We thank the reviewer for his/her knowledgeable summary and positive comments.

Comment 1.2

I have a few comments, which I am sure the authors can be easily address:

1)In this reviewr’s opinion, the “large zoo of irreps” cannot be entirely left the Appendix, and more substance needs to be added to Eq. 2.

2)Even in the appendix where the irreps are discussed, the presentation is not so clear. For example, the summary in Ref. [20] at pages 151-152 (in the 2009 print of the book) on how ALL irreducible reps of a space group is constructed should be included in one form or another.

3)Some of the conclusions in D2 about the physical content are important and should be presented in the main text.

Reply 1.2 We thank the reviewer for the suggestions. We have amended the manuscript as advised. (List of changes items 13, 14, and 16.)

Comment 1.3

4)Since the BS is defined by a gap condition over the highly-symmetric points of the Brillouin zone (only), the present analysis allows for gap-closings at regular k-points. As such, the topological phases identified this way can correspond to either insulator or semi-metals. Can

these two cases be differentiated by arguments similar to those related to Eq. A3? More precisely, for the metallic case one should find that the representations cannot be extended continuously over the whole Brillouin zone (hence they must be connected to other bands).

Reply 1.3 The two cases can indeed be differentiated, at least for specific symmetry settings. In brief, we need to first divide the (semi-)metallic systems with gap closing at non-high-symmetry momenta into two classes: (i) The (topologically-protected) gap closings can be annihilated with their counter parts without affecting the band gap at any high-symmetry momentum; and (ii) The gap closings cannot be removed as long as the symmetry representation content at the high-symmetry momenta is held fixed. The representation content for case (i) is compatible with a band insulator, whereas that for (ii) is only compatible with (semi-)metals. We refer to case (ii) as representation-enforced semimetals (reSM). Therefore, if we can tell case (i) from (ii), the question is answered. This is partially achieved in Supplementary Note 5, where we analyze the stability of Weyl points, and discuss how the reSM classification is related to $X_{\{BS\}}$. Note that, in that discussion, we have focused on 3D, spinful systems without time-reversal symmetry. In other symmetry setting, nodal lines (say) can also be topologically stable, and we defer the full analysis of these problems to future works.

Comment 1.4

5)It is unfortunate that some existing and highly regarded works from mathematical physics literature have been not cited. Some of these works are:

Ken Shiozaki, Masatoshi Sato, Kiyonori Gomi, “Topological Crystalline Materials - General Formulation, Module Structure, and Wallpaper Groups” arXiv:1701.08725 (2017)

Kiyonori Gomi, “Twists on the Torus Equivariant under the 2-Dimensional Crystallographic Point Groups”, SIGMA 13, 014 (2017).

Ken Shiozaki, Masatoshi Sato, Kiyonori Gomi, “Z2 topology in nonsymmorphic crystalline insulators: Möbius twist in surface states”, PRB 91, 155120 (2015).

Ken Shiozaki, Masatoshi Sato, Kiyonori Gomi, “Topology of nonsymmorphic crystalline insulators and superconductors”, PRB 93, 195413 (2016)

Reply 1.4 The mentioned references have been added. (List of changes item 19)

Reviewer #2

Comment 2.1

Review on “Complete Theory of Symmetry-based Indicators of Band Topology” by Hoi Chun Po, Ashvin Vishwanath and Haruki Watanabe.

I have read the manuscript as well as the refs [13] and [23] in the paper with care. In essence the subject of new classification procedures and associated physical consequences in presence of

space group symmetries is rather timely. Not in the least place due to the recent work, refs [13] and [23], and work by the Princeton group.

The authors exploit a recent connection between K-theory and combinatorial constraints from ref [13] and their previous work on the essential role of the filling constraint, ref [23]. All in all, this means that the bands have to be glued together in a symmetry respecting fashion, connecting high symmetry points in a certain number of possible manners. Taking into account filling constraints one can determine the system and also point out the different “topological possibilities”, in the view of the paper. The extended procedure by the authors then conspires in a somewhat related framework. That is, the band structure is essentially determined by the kernel of the matrix of constraints [after lifting it to integers instead of non negative integers], Eq. 1. This structure thus reveals a natural abelian group, their $Z^{d_{BS}}$. Most importantly, the authors then extend and contrast these insights with the Atomic, that is trivial, insulator (AI). This reveals that the topological variants can be seen as a “fraction” of the AIs and results in Eq. 5.

Apart from technicalities this essentially pertains to rather heuristic insights. Indeed physically it boils down to the fact that for an AI one can find a localized states, the Wannier states, whereas for the topological counter parts this had been long known not to be possible. These “quantum band insulators” (QBIs) arising by the mismatch thus, in very crude essence, form the class of other states, such as topological insulators and relate to the integers/constraints as described. With these insights in hand the authors also demonstrate new insulating cases and semimetallic cases falling within this paradigm.

I find the paper interesting and the presentation very clear apart from some minor issues. The question is thus whether the content meets the standards of Nature Communications. The framework is interesting, fairly general and clear, but also very much an extension of the insights of the previous refs indicated. Nonetheless the authors also propose a new quantum band insulator and the lattice-enforced semimetal. Problem though is that, focussing on the first one for example, the physical effects are not clear; the new quantum band insulator has no protected surface states and a trivial magnetoelectric response. From a some viewpoints one might thus question whether we are not dealing with a nomenclature for band theory. On the other hand, from the entanglement spectrum one does see a difference.

Taking these thoughts into account I am inclined towards endorsing publication in combination with the incorporation of some suggestions.

Reply 2.1 We thank the reviewer for his/her knowledgeable summary and positive comments. While we agree that the present work is built on the insight that band structures incompatible with any real-space description are necessarily topological, as was discussed earlier in Refs. [13,14,17,29,30], we stress that we have provided the first systematic, readily-computable framework that is applicable to all space groups with or without spin-orbit coupling and time-reversal symmetry. Besides, we also note that it is too early to assert that quantum band insulators without any protected surface states have no nontrivial physical properties—as was discussed in Ref. [13], one quite generally expects the possibility of finding quantized physical responses associated to these non-atomic band structures. Whether or not such quantized

responses exist for every symmetry setting is an important open problem. In addition, we do not think that the new phases we discussed are the only highlights of the paper: the simple framework we developed allows one to analyze the symmetry properties of any band structures, gapped/ gapless and topological/ trivial, in an efficient manner. The realization that any band structure can be expanded in terms of an “atomic basis”, and that this expansion encodes important physical properties, is a fundamental aspect of the band theory which we discovered.

Comment 2.2

-As said the new phases are the highlight of the paper. These have therefore to be explained a bit better in my opinion. I understand the motivation, but the crucial difference, the entanglement spectrum of the 'doubled' strong TI should then be shown in the figure.

-More generally the figures are essential part of the paper, in the text it is often referred to as an explanation. Also many readers may focus on the figures. Therefore the captions have to be improved a lot. They should be self-contained. Eg. in fig 1, the a and b panel can be linked more to explain better that the ν and constraints are linked in a particular fashion. Maybe the authors want to interchange here the a and b panel. Also in figs 2 b and c the left part is unclear. I get what they are saying from the text, but in the figure itself it is rather obscure what the authors like to convey. For 2c one just sees an insulating band structure. It is known that these can be different from the AIs. So explain this with a understandable cartoon, I presume the left part, or leave it out.

-Also the captions of the tables should be expanded. Readers skimming through the text should be guided what the tables convey.

-On page 6 the authors consider the spin quantum numbers $1/2 \oplus 3/2$ these are for J. So please name it to make it more clear. Again either attach all the labels or just name the multiplets in words.

Reply 2.2 We thank the reviewer for the suggestions. We have made corresponding changes in the revised manuscript. (List of changes items 4,5,9,10,11,& 12)

Comment 2.3

*-Having read ref 23, I think the authors should change
“ For instance, in Ref. [23], which focuses on systems in the 17 2D wallpaper groups without any additional symmetry, such an approach was adopted to help develop a more physical understanding of the mathematical treatment of Ref. [24]”*

to

” For instance, in Ref. [23], which focuses on systems in the wallpaper groups without any additional symmetry, such an approach was adopted to help develop a more physical understanding of the mathematical treatment of Ref. [24]”

on page 1. The method is focussed indeed on 2d but in fact general.

Reply 2.3 We have amended the text as suggested. (List of changes item 2)

Comment 2.4

-The authors should remove complete from the title. The precise connection to the mathematical structure is not yet completely identified. Also things like half mirror Chern numbers are not always picked up by these relations and should therefore maybe be re-evaluated.

Reply 2.4 We would like to clarify the reason for our title "Complete Theory of Symmetry Indicators of Band Topology." As the referee has pointed out, in this work we do not attempt to develop a "Complete Theory of Band Topology." Other works such as Ref. [22] have attempted to do this. Our goal instead focused on *symmetry indicators* of band topology. This is a rather different enterprise as we have emphasized. For example, there are symmetry settings for which topological phases are not detectable using symmetries alone, as in the case of any topological band insulators in SG1, the space group with only lattice translation symmetries. The task of determining the Z-valued (mirror) Chern numbers also falls under the same category. However, we do believe our theory is complete when it comes to symmetry indicators of band topology. That is, *all* possible indicators for *all* space groups have been computed using a single, general framework. We would therefore like to stand by our original title. Nonetheless, in response to the referee's comment we have added additional discussions to clarify this point. (List of changes item 15.)

Reviewer #3

Comment 3.1

This manuscript presents a mathematical framework allowing to identify topological band structures in solids. More precisely its ambition is to define a theory which allows to explore all space groups of solids and identify topological properties tightly bound to these crystalline symmetries. The tools and the method for this task are defined in this paper, while the exhaustive search is postponed to future work. As such, this manuscript is naturally rather abstract and focuses more on methodology than on physical result. From this perspective, while presenting new material, it is related to several recent papers, such as references [24] and [23]. Overall, I have found this manuscript rather clear and pedagogical. The core of this paper corresponds to a discussion of the method, while all technical details are postponed to the supplementary materials (and this one is very dense). However, in a paper which already requires some special attention due to its technical nature, the frequent use of acronyms really hampers the reading. The final list is rather long (TR, SOC, QBI, BS, AI, feQBI, reQBI, reSM, DOF, leSM, feSM,...). Notwithstanding this annoyance, the reading of this manuscript was pleasant and very interesting.

In this end, my opinion is that this manuscript is a very formal one. However its subject and results are of high importance in the recent domain of topological phases. The experimental search for topological phases has strongly relied on formulation of topological invariants

restricted to the presence of special symmetries such as inversion, or crystalline symmetries for semi-metals. Nevertheless no general framework starting from these symmetries existed prior to this paper. It is my understanding that such a framework will be of very high relevance for the search of new topological materials of various nature, and that this work should be published in Nature Communications.

Reply 3.1 We thank the reviewer for his/her knowledgeable summary and positive comments. We have also tried to improve the readability of the paper by reducing the acronyms involved, taking away some non-standard ones like SOC and DOF. We have also added a summary of the abbreviations under the Methods section. (List of changes item 8)

Comment 3.2

Finally let, me had a minor comment concerning the "applications" section of this article, which I find not illustrative enough. In particular, I wonder whether part of the discussion of sec. VI of the supplementary should be included on the main text.

Reply 3.2 We have moved the original discussion on lattice-enforced semimetals in the supplementary materials to the Methods section of the main text. (List of changes items 5 & 6)

List of changes

1. Abstract reorganized to comply with formatting requirements
2. Discussion on (updated) Ref. [28] in the introduction is revised.
3. Results on quasi 1D and 2D systems are incorporated, reflected in a brief mentioning in the 2nd paragraph of “Overview of strategy and results”, a subsection “Extension to other symmetry settings” in methods, and the referencing to Supplementary Tables 9-20
4. We expanded the discussion in the 3rd and 4th paragraphs of the subsection “Quantum band insulators in conventional settings” to elaborate on the entanglement signature of the doubled strong TI phase
5. Discussion on the details of the leSM model in the 3rd and 4th paragraphs in the subsection “Lattice-enforced semimetals” are simplified for clarity
6. Original subsection on example of “filling-enforced quantum band insulators in unconventional settings” is moved to Supplementary Note 4. The associated figure (sub panel in Fig. 2) is correspondingly moved.
7. At the end of the Discussion section, we added a paragraph discussing relation of the present work and a recent preprint that appeared (Ref. [40])
8. Abbreviations “SOC” and “DOF” are removed; under methods, we added a “Glossary of abbreviations”
9. We also moved the original supplementary materials concerning the details of the leSM example to the methods section. In this discussion, we clarified that the spin quantum numbers refer to the total spin.
10. Figure and Table captions significantly expanded
11. Ordering of the two panels in Fig. 1 swapped
12. Fig. 2(d) is revised to provide more details
13. We added a subsection under Methods to discuss how the ideas in the 1D example apply to the 3D case
14. A discussion on the physical aspects of {BS} and X_{BS} are added under Methods
15. Elaborated on the relation to K-theory classifications in Methods
16. Discussion in Supplementary Note 1 concerning the construction of irreps is expanded
17. Other formatting changes throughout the manuscript to comply with editorial requirements, clarify the discussions, and some typos fixed
18. Added grant information for H.W.
19. The following references have been added:
 - [20] Shiozaki, K., Sato, M. & Gomi, K. Z₂ topology in nonsymmorphic crystalline insulators: Mobius twist in surface states. Phys. Rev. B 91, 155120 (2015).
 - [21] Shiozaki, K., Sato, M. & Gomi, K. Topology of nonsymmorphic crystalline insulators and superconductors. Phys. Rev. B 93, 195413 (2016).
 - [23] Gomi, K. Twists on the Torus Equivariant under the 2-Dimensional Crystallographic Point Groups. SIGMA13, 014 (2017).
 - [24] Shiozaki, K., Sato, M. & Gomi, K. Topological Crystalline Materials - General Formulation, Module Structure, and Wallpaper Groups-. Preprint at <https://arxiv.org/abs/1701.08725> (2017).

- [29] Soluyanov, A. A. & Vanderbilt, D. Wannier representation of \mathbb{Z}_2 topological insulators. Phys. Rev. B 83, 035108 (2011).
- [40] Bradlyn, B. et al. Topological quantum chemistry. Preprint at <https://arxiv.org/abs/1703.02050> (2017).
- [41] Kopský, V. & Litvin, D. B., ed., International Tables for Crystallography. (Wiley) Vol. E: Subperiodic groups, 2nd edition (2010).